# From Earlier Exploration to Advanced Applications: Bibliometric and Systematic Review of Augmented Reality in the Tourism Industry (2002–2022)

Mohamed Zaifri [1],* , Hamza Khalloufi [1] , Fatima Zahra Kaghat [2], Ahmed Azough [2] and Khalid Alaoui Zidani [1]

[1]  LISAC Laboratory, Faculty of Sciences Dhar El Mehraz, Sidi Mohamed Ben Abdellah University, Fez 30003, Morocco
[2]  Research Center, Léonard de Vinci Pôle Universitaire, 92916 Paris, France
*   Correspondence: mohamed.zaifri1@usmba.ac.ma

**Abstract:** Augmented reality has emerged as a transformative technology, with the potential to revolutionize the tourism industry. Nonetheless, there is a scarcity of studies tracing the progression of AR and its application in tourism, from early exploration to recent advancements. This study aims to provide a comprehensive overview of the evolution, contexts, and design elements of AR in tourism over the period (2002–2022), offering insights for further progress in this domain. Employing a dual-method approach, a bibliometric analysis was conducted on 861 articles collected from the Scopus and Web of Science databases, to investigate the evolution of AR research over time and across countries, and to identify the main contexts of the utilization of AR in tourism. In the second part of our study, a systematic content analysis was conducted, focusing on a subset of 57 selected studies that specifically employed AR systems in various tourism situations. Through this analysis, the most commonly utilized AR design components, such as tracking systems, AR devices, tourism settings, and virtual content were summarized. Furthermore, we explored how these components were integrated to enhance the overall tourism experience. The findings reveal a growing trend in research production, led by Europe and Asia. Key contexts of AR applications in tourism encompass cultural heritage, mobile AR, and smart tourism, with emerging topics such as artificial intelligence (AI), big data, and COVID-19. Frequently used AR design components comprise mobile devices, marker-less tracking systems, outdoor environments, and visual overlays. Future research could involve optimizing AR experiences for users with disabilities, supporting multicultural experiences, integrating AI with big data, fostering sustainability, and remote virtual tourism. This study contributes to the ongoing discourse on the role of AR in shaping the future of tourism in the post COVID-19 era, by providing valuable insights for researchers, practitioners, and policymakers in the tourism industry.

**Keywords:** augmented reality; immersive technology; tourism; bibliometric analysis; content analysis; mobile AR; COVID-19; future trends; AR design; emerging topics

## 1. Introduction

In recent years, immersive technologies have become increasingly popular in various industries, including tourism. Augmented reality (AR) and virtual reality (VR) are being implemented in various tourist and hospitality areas, such as theme parks, museums, historical sites, etc. [1]. These technologies have gained significant attention from researchers and practitioners, due to their capability to enhance tourists' satisfaction, by delivering unforgettable experiences [2].

AR technology, a variant of virtual environments (VE) that allows users to see the real world with virtual objects overlaid in real time [3], has emerged as a transformative tool that can revolutionize various sectors, offering enhanced and immersive experiences. Researchers and tourism practitioners have recognized AR's potential in tourism since 2000 [4]. Pioneering studies, such as that by Vlahakis et al. [5], have introduced novel

systems that offer personalized AR tours and 3D reconstructions of ruined historical sites. Additionally, advancements in AR technology, as highlighted by Höllerer and Feiner in their study [6], have enabled tourists to easily discover destinations, access background information, and explore countless possibilities, limited only by the capabilities of the AR device, and the available information. Since then, AR has been increasingly adopted to provide tourists with relevant information about the sites they visit, improve navigation, and create highly dynamic experiences at tourist attractions [7]. The growing interest in AR applications for tourism requires a comprehensive understanding of AR technology's evolution, contexts, and design elements in this field.

Moreover, tourism is facing numerous challenges related to technology, economics, and sustainability [8]. The COVID-19 pandemic has further intensified these challenges, causing significant socio-cultural and economic impacts on various stakeholders in the industry, with some effects expected to persist for years to come [9]. In addition, with the rapid development of mobile technology, and the emergence of groundbreaking AR applications, such as the Pokémon Go craze [10], tourists are showing an increasing demand for AR-based solutions to enhance their travel experiences [11].

By providing a comprehensive overview of AR research in tourism, this study intends to inform researchers, industry practitioners, policymakers, and educators about the current state of the field, potential research directions, and practical implications. Ultimately, this knowledge can contribute to the advancement of AR technology in tourism, improving the overall tourist experience, and relaunching tourism post-COVID-19.

While most studies focus on conducting reviews that often encompass various immersive technologies in tourism (i.e., AR, VR, MR), this study's contribution to the literature is to conduct a comprehensive review on the use of AR specifically, in the context of tourism. This review covers a twenty-one-year period (2002–2022), and utilizes a dual-method approach. Firstly, a bibliometric analysis was conducted on 861 articles to (1) highlight emerging topics and contexts of AR use in tourism, and (2) explore the evolution of research production over time and across countries. Secondly, we carried out a systematic content analysis on the 57 selected studies, to (3) summarize the most commonly used AR design elements in tourism settings. Finally, this study offers insights into research gaps and potential future directions. Consequently, the following research questions are addressed:

RQ1. How are research articles on AR in tourism distributed temporally and regionally over the period (2002–2022)?

RQ2. What are the main contexts in which AR research was used to support the tourism industry, and how has this evolved over time?

RQ3. What are the most commonly used AR design components (systems, devices, virtual content, tourism settings), and how do they integrate to enhance the tourism experience?

This article is structured as follows. Section 2 discusses related work from previous reviews, and the background of this study. Section 3 outlines the methodology employed in the review. Section 4 illustrates and discusses the findings, including emerging topics and contexts of AR use in tourism, research production over time and across countries, and commonly used AR design elements in tourism environments. Section 5 explores future research directions for AR in tourism. Finally, Section 6 provides a conclusion to the article, including a summary of the key findings, limitations of the study, and theoretical and practical implications.

## 2. Previous Work

In recent years, there has been a growing interest in immersive technologies in tourism research, leading to numerous published review studies. In this section, we present and discuss some of the most recent reviews, to identify trends and patterns in the literature.

Boboc et al. [12] published a recent bibliometric review of 1201 articles over the past decade (2012–2021), which revealed eight trending topics of AR application to cultural heritage (CH), such as virtual museums, gamification, e-heritage, and user experience. Ideas on existing applications were described, based on discussing each trending topic.

Jingen Liang and Elliot [4] provided a systematic review focusing on AR in the tourism literature. The results identified five emergent clusters, and established a robust statistical relationship between the perceived ease of use and the perceived usefulness for behavioral intention, which could provide reasonable evidence for future research alongside AR gamification.

Wei [1] presented a critical review of the progress of AR and VR research in tourism and hospitality, wherein they synthesized the stimuli, dimensions, and consequences of AR/VR related to user behavior, and highlighted fruitful directions for future research, such as the need for a cross-cultural approach, and predictive research on technological advances.

In a review conducted by Yung and Khoo-Lattimore in 2019 [13], seven categories were identified within which the use of AR and VR in tourism has emerged, including marketing, experience enhancement, and tourism education. The review also identified gaps and challenges in the field, such as the usability and low awareness of AR and VR, and concluded that further theory-based research was needed to advance the field.

Loureiro et al. [2] reviewed current emerging topics, and avenues for future direction of AR and VR in tourism between 1995 and 2019, concluding that tourism experiences would be enhanced by adopting more brain–computer interfaces, wearable devices, and physical stimulation. The results also showed significant similarity among the emerging topics in the conference papers and journal articles.

Fan et al. [14] introduced an innovative meta-analytic framework that differs from previous review studies by examining the influence of the operating mechanism of AR/VR tourism applications on enhancing the tourist experience. This framework integrates both AR and VR technologies, and aims to identify the essential characteristics and theoretical operating mechanisms of immersive technologies in the context of tourism. The developed meta-analytical framework includes 24 constructs, derived from 65 independent studies in 56 articles, drawing upon the body of empirical literature mentioned in [2]. The analysis findings, involving 472 relationships, emphasize the pivotal role of "presence" as a key feature of AR/VR in the tourism domain. Additionally, the study reveals the positive moderating effects of "simulation type" and "social interaction" on the impact of presence on the tourism experience, while "prior visitation" demonstrates a negative moderating effect. Fan et al. justified the integration of the AR and VR analyses by stating that AR is a particular form of VR.

Zhou et al. [15] conducted a recent review on the use of AR and VR in museum education. They performed a meta-analysis of 17 studies, to evaluate the impact of these technologies on learning outcomes.

Bekele et al. [16] presented a comprehensive overview of the current status of AR, VR, and MR technologies in the CH domain, emphasizing the restrictions of present technologies, and the requirement for further research. The study also presented a framework for evaluating different systems, and determining the most appropriate solutions for a specific application.

In a broader context, Manuri and Sanna [17] conducted a survey that explored the primary application domains of AR, providing an overview of current technologies and future trends. Their study identified various domains, including tourism and cultural heritage, as well as education, medicine, the military, and entertainment. The authors concluded that the progress of AR technologies is closely linked to advancements in AR devices, and the availability of content, which will be further discussed in the systematic review section of our study.

It was noted that there is a growing trend in the literature for reviews that cover both AR and VR technologies. However, it is important to distinguish between these technologies, as they have unique characteristics and purposes. AR enhances the user's perception of the real world, by overlaying virtual information, while VR creates a fully immersive and artificial environment for the user experience. As reported by Yung and Khoo-Lattimore [13], the cluster of studies related to enhancing the tourism experience using emerging VR and AR technologies in tourism was found to be exclusively composed

of AR studies. This highlights the need for further research into the specific application of AR in tourism.

The above-mentioned studies also indicate that the use of a combination of qualitative and quantitative methods of analysis in literature reviews is uncommon. Bibliometric analysis provides a quantitative perspective on the literature, including identifying trends and highly cited works [12,18,19], while content analysis offers a qualitative perspective, including research methodologies, gaps, and implications [1,15]. A more holistic and nuanced understanding of AR research within the field of tourism could be achieved by incorporating both bibliometric and content analyses.

To the best of the author's knowledge, this is one of the earliest studies to employ a dual-method approach, combining bibliometric and systematic review techniques, to explore the past, present, and future impacts of AR technology on the tourism industry.

## 3. Materials and Methods

The bibliometric methodology involves applying quantitative techniques to bibliographic data, allowing researchers to unravel the evolving nuances of a specific field, while highlighting emerging areas [20]. In the present study, we conducted the bibliometric analysis to identify the key domains and contexts within the tourism industry that have been influenced by AR (RQ2). Additionally, the temporal evolution and geographical distribution of the research were determined through the application of bibliometric analysis (RQ1).

Systematic content analysis, on the other hand, is a commonly utilized research method that involves the subjective interpretation of textual data content through a systematic process of classification, based on coding and theme identification [21]. In this research paper, we employed content analysis to identify and summarize the design elements of AR technology for tourism applications (RQ3). Later, future directions of AR in tourism are discussed, based on the current study findings.

### 3.1. Search Strategy and Data Collection

The relevant literature on AR in the tourism industry was acquired from the Scopus and Web of Science (WoS) online databases, using a search query applied to the text, abstract, and keywords. The search was performed on 23 January 2023, using the following query: (("*augmented reality*") AND ("*touris*\**")) with a date range from 1 January 2002 to 31 December 2022.

Scopus and WoS are the two primary existing multidisciplinary databases for obtaining as many relevant and quality articles as possible, supported by previous studies and reviews, and found to be sufficiently compelling [2,4,13,15]. For both Scopus and WoS, several restrictions were imposed to obtain the final results. The document type was restricted to articles and conference papers, and only final articles published in peer-reviewed journals, or conference proceedings in English were retained.

The initial database search returned 1206 articles, with 615 sourced from Scopus and 591 from WoS. These records were then imported into the literature management tool "Zotero" for duplicate removal, and subsequent literature selection for systematic content analysis. After merging both datasets in the "Zotero" tool, 345 duplicates were eliminated, leaving 861 articles for further screening and selection. Later, the 861 records were exported to the Bibliometrix R tool, "Biblioshiny", for comprehensive bibliometric analysis.

### 3.2. Literature Screening and Selection Process

The PRISMA 2020 framework [22] guided the systematic process of literature identification and screening, as illustrated in Figure 1. After the removal of duplicates, 861 records were left for further filtering based on the selection criteria described in Table 1. Initially, a preliminary screening was performed on the title and abstract of each record, resulting in the exclusion of 583 records that did not meet the inclusion criteria, with the exception of the second criterion. Of the remaining 278 records, we succeeded in retrieving the full texts

of 118 reports for analysis. Despite our concerted efforts, which included a combination of manual (online search) and automated (Zotero) methods, we were unable to overcome the limitations that hindered our access to the additional 160 records, including restricted access, limited availability, and/or technical constraints. Subsequently, the authors performed an in-depth analysis of the full text of each report against inclusion and exclusion criteria, to uncover any discussions that were not reflected in the abstract. Ultimately, 61 studies were removed; 15 of them not describe any implementation of AR systems, 1 combined AR and VR technologies, 5 were missing sections of the article, 18 reported pure theoretical research on AR in tourism, and 22 lacked evidence of the application of AR in tourism settings. Despite searching for additional records through citations, no further articles were found. Thus, the final number of articles included in the review for content analysis was 57 (see Supplementary materials: Table S1). Two authors reviewed all articles during the entire selection process, and resolved any discrepancies between their findings through discussions, until a consensus was reached.

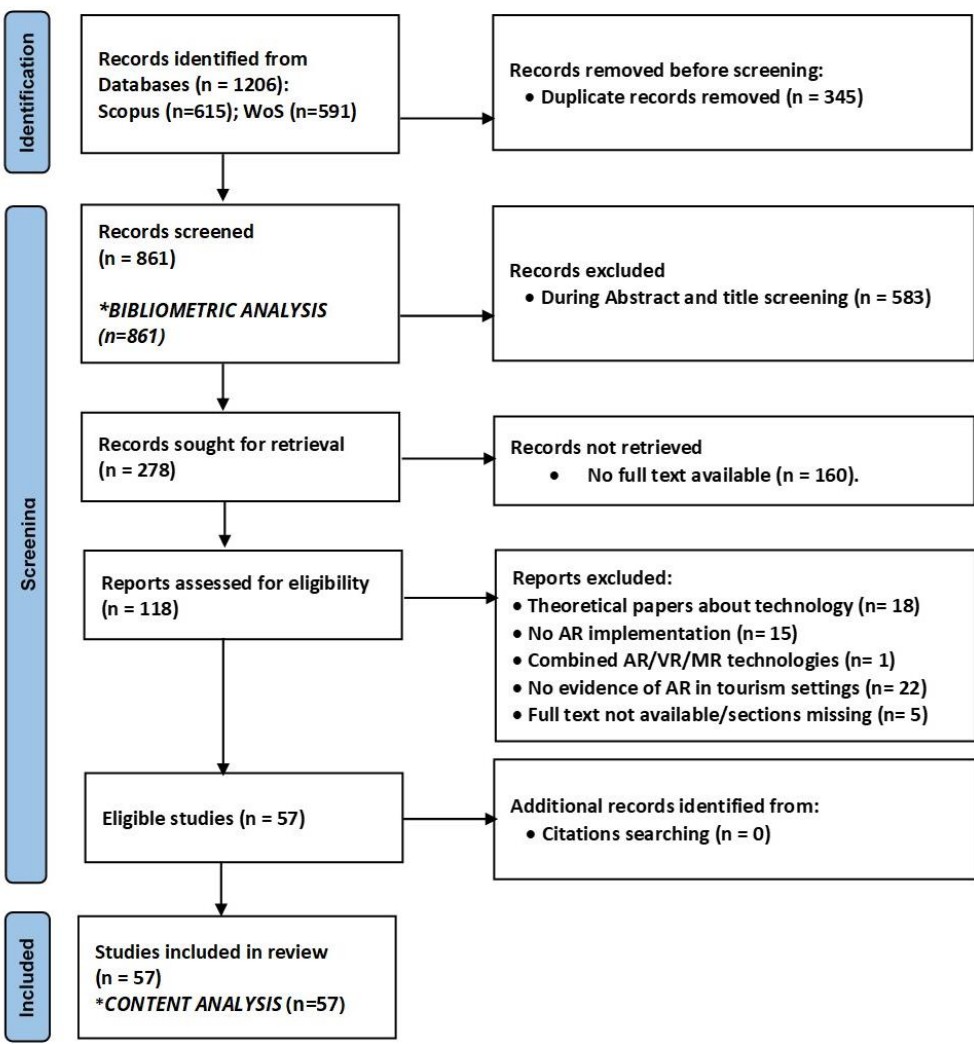

**Figure 1.** Process of selecting final papers guided by PRISMA 2020 framework.

**Table 1.** Inclusion and exclusion criteria for data selection.

| Criteria | Inclusion | Exclusion |
|---|---|---|
| Language | English | Not English |
| Access | Full text | Full text not available, missing sections, preprints |
| Type of article | Journal article, conference paper, book chapter | Poster, editorial |
| Research methodology | Empirical study | Theoretical study, review, extended abstract |
| Evidence of AR in tourism | Reported | Not reported |
| Addressing RQ3 | Yes | No |
| Field of study | Pure AR | Combined AR/VR/MR |

*3.3. Data Analysis*

3.3.1. Bibliometric Analysis

In the present study, we employed two main methods of bibliometric analysis: performance analysis, and scientific mapping. Performance analysis involves the use of metrics to assess the contributions of research elements (e.g., authors, countries). Total publications (TP) and total citations (TC) were used as metrics to gain valuable insights into the productivity and impact of countries on AR research in tourism, which helped to address our first research question (RQ1).

Science mapping, on the other hand, involves the exploration of relationships between research elements. In this study, we used co-word analysis (i.e., word co-occurrence) to identify the main areas of AR application in tourism, and to depict the conceptual structure of emerging topics. Specifically, we conducted a keyword co-occurrence analysis, using the authors' keywords to identify the main themes that emerged from the research literature on AR in tourism, thereby answering our second research question (RQ2).

To visualize and analyze the bibliographic data in our study, we employed Bibliometrix, an open-source tool that utilizes R programming for scientific mapping analysis. This tool enabled us to generate networks and visualizations of the data. The use of Bibliometrix is a recognized and widely used method for scientific mapping, and has been shown to be a significantly flexible and time-efficient tool for network generation [23].

3.3.2. Content Analysis and Coding Framework

To answer RQ3, the content analysis technique was applied using a directed approach [21]. An initial coding framework was established, based on earlier reviews, to categorize the selected studies and summarize the findings of the 57 articles, according to various dimensions, including AR tracking systems, AR technology devices, virtual content overlaid, and tourism settings. Through this analytical approach, sub-themes emerged, and their characteristics were established.

- Codes for AR systems:

Based on the classification of immersive technologies types proposed by Kuhail et al. [24], this study coded the AR tracking systems into three categories, with five subcategories, as depicted in Table 2.

**Table 2.** Coding framework for AR tracking systems.

| Category | Sub-Category | Explanation |
|---|---|---|
| Marker-based | Marker-based image | AR technology uses predefined images (e.g., QR codes) to track and overlay virtual content. |
| | Marker-based object | AR technology uses predefined physical 3D objects (e.g., chair, table) to track and overlay virtual content. |
| Marker-less | Location-based | AR technology takes a physical location in the form of GPS coordinates, and delivers customized digital content based on that location. |
| | Projection-based | Video projection technology is used in combination with AR technology, to enhance physical environments by overlaying virtual images onto physical surfaces (e.g., a virtual keyboard displayed on a table that enables users to input data). |
| | Superimposition-based | AR technology overlays digital content directly onto real-world objects in real time (e.g., tracks a person's facial features and adds virtual glasses). |
| Hybrid | - | Hybrid systems utilize a combination of technologies from both marker-based and marker-less systems, including projects that utilize different technologies within the same system. |

- Codes for tourism settings:

Following the work published by Bekele et al. [16], the study classified tourism settings into four categories, which are represented in Table 3.

**Table 3.** Coding framework for tourism settings.

| Category | Explanation |
|---|---|
| Indoor | Refers to AR experiences in closed environments (e.g., museums, exhibitions, shopping centers). |
| Outdoor | Refers to AR experiences in open environments (e.g., natural parks, cities, cultural landmarks). |
| Combined | Refers to AR experiences in both open and closed environments. |
| Not specified | The location of the AR experience is not specified, either because it is not relevant to the study, or because it is not mentioned. |

- Codes for AR devices:

The codes for AR technology devices are given in Table 4, and follow the coding frameworks presented in the works of Bekele et al. [16] and Zhou et al. [15].

**Table 4.** Coding framework for AR devices.

| Category | Explanation |
|---|---|
| Handheld | AR technology is deployed on mobile and wireless computing devices (e.g., smartphones, tablets). |
| Head-Mounted-Display (HMD) | AR technology is deployed with head-worn devices or built-in helmets that contain a small computer and monitor (e.g., AR Smart Glasses, AR headsets). |
| Desktop | AR technology is deployed using stationary desktop computers and digital devices that cannot be easily relocated (e.g., AR training simulator). |
| Spatial | Augmentation is achieved using video projectors and tracking devices, to project digital content directly on the physical surface. |
| Not specified | AR device was not specified in the study. |

- Codes for virtual content:

In this study, the virtual content used by AR technologies has been categorized into six distinct types based on the senses they stimulate, as listed in Table 5.

**Table 5.** Coding framework for virtual content.

| Category | Explanation |
|---|---|
| Visual | Images, graphics, and videos are used to enhance the visual experience, such as 2D objects or 3D models. |
| Auditory | The sound is used to enhance the AR experience (e.g., background music, sound effects, voice recordings). |
| Haptic | Touch sensations are used to add a sense of physical interaction to the AR experience (e.g., vibration, force feedback). |
| Olfactory | Scents are used to enhance the AR experience. This type of virtual content can be used by adding familiar scents to the environment. |
| Gustatory | Gustatory virtual content uses taste to enhance the AR experience. This type of content can be used to simulate the taste of food or drinks through haptic technology or other means. |
| Multisensory | Multisensory virtual content combines multiple senses, including visual, haptic, auditory, olfactory, and gustatory, to create a fully immersive AR experience. |

The initial stage of developing the coding frameworks involved discussions between the first two authors, with input and approval from the other authors. In cases of disagreement, further discussions were held, until a consensus was reached.

## 4. Findings and Discussion

Based on the bibliometric analysis, over the course of two decades (2002–2022), a total of 861 primary research articles focused on the application of augmented reality (AR) within the tourism sector were documented by the principal databases of Scopus and WoS. These publications demonstrated an annual growth rate of 25.77%, indicating a steady increase in interest in this field over time. The compilation of articles originated from 610 distinct sources, with 1944 Plus keywords, and 2105 author-specific keywords. The authorship of these articles involved 2463 authors, with 79 single-authored publications, and the remaining articles averaging 3 to 4 authors per paper. The rate of international collaboration among authors was 13.43%, indicating a considerable interest in global partnerships among scholars in the area of AR applied to tourism.

### 4.1. Annual and Regional Distribution of Scientific Production

The growth of research in the field of AR for tourism can be assessed by analyzing the annual distribution of published papers. The timeline of the number of research papers published per year is demonstrated in Figure 2, which reveals three distinct periods of productivity, each characterized by specific underlying factors.

The first period, spanning from 2002 to 2008, marks the nascent stage of AR tourism research, with a minimal output of 1 to 5 publications per year. The limited productivity can be attributed to the novelty of the technology, the scarcity of practical applications, and the lack of widespread awareness about the potential benefits of AR in tourism. This period is characterized by the pioneering efforts of researchers to investigate the potential of AR technology in the tourism industry.

The second period, covering 2009 to 2019, exhibits a steady increase in research output, with annual publication numbers ranging from 8 to 131. This growth can be attributed to the maturation of AR technology, increased accessibility, and a growing recognition of its potential application in the tourism sector. During this period, researchers began to explore a wider range of topics, such as design, development, gamification, user experience, satisfaction, and behavior intention, demonstrating an expanding interest in the field (see Section 4.2).

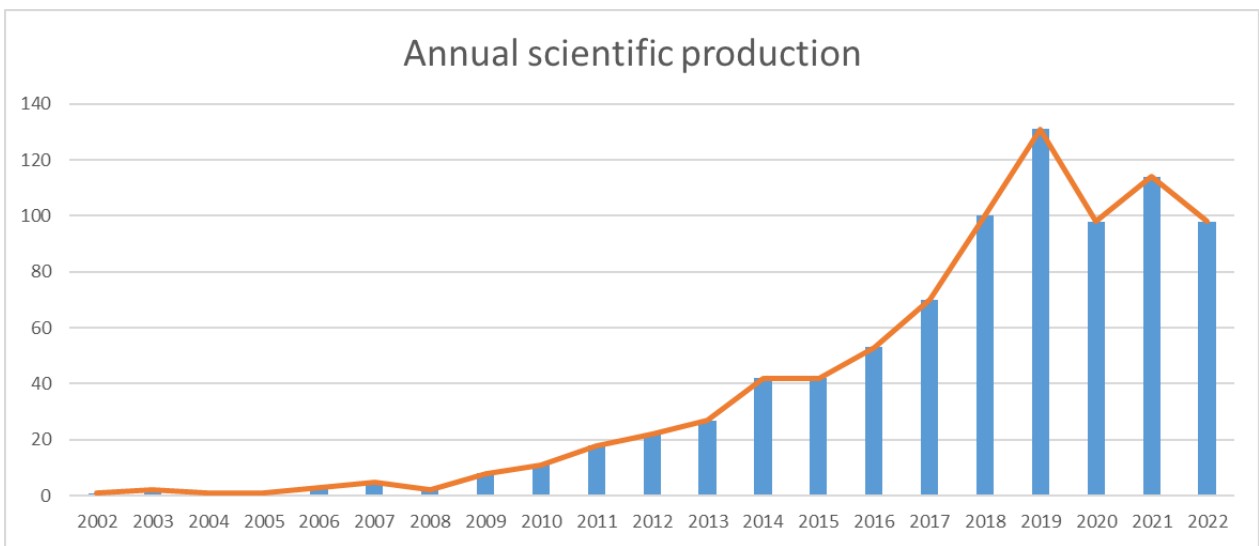

**Figure 2.** Annual scientific production.

The third period, extending from 2020 to 2022, features consistent annual publication numbers, stabilizing at around 98 in both 2020 and 2022. This period, known as the post-COVID-19 era, commences in 2021, after a decline in research productivity during 2020 due to the pandemic. This phase is characterized by a renewed interest in AR tourism research, as the tourism industry pursued innovative approaches, such as artificial intelligence and machine learning, to adapt to novel challenges and shifting consumer behaviors resulting from the pandemic.

Figure 3 demonstrates the world map of regional scientific production. Delving deeper into the productivity of countries in the realm of AR tourism research, Italy emerges as a frontrunner, with an impressive 170 publications (20% of the total publications), followed closely by China at 142 (17%), and the UK with 87 (10%). When juxtaposed, these countries exemplify the varied regional and economic contexts that contribute to the field's diversity and growth.

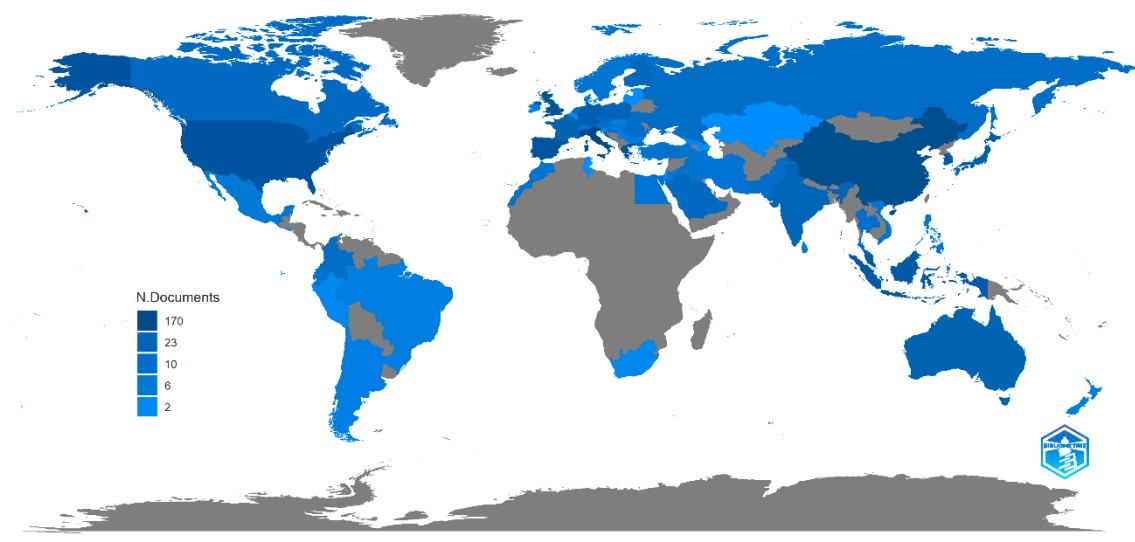

**Figure 3.** Geographical distribution of scientific production.

As shown in Figure 4, Europe is strongly involved, with countries such as Italy, the UK, Greece, Portugal, and Spain contributing significantly to AR tourism research. Concurrently, Asian nations such as China, Malaysia, South Korea, and Indonesia reveal considerable dedication to the field. The United States of America (USA), as the leading contributor from North America, further highlights the global scope of AR tourism research.

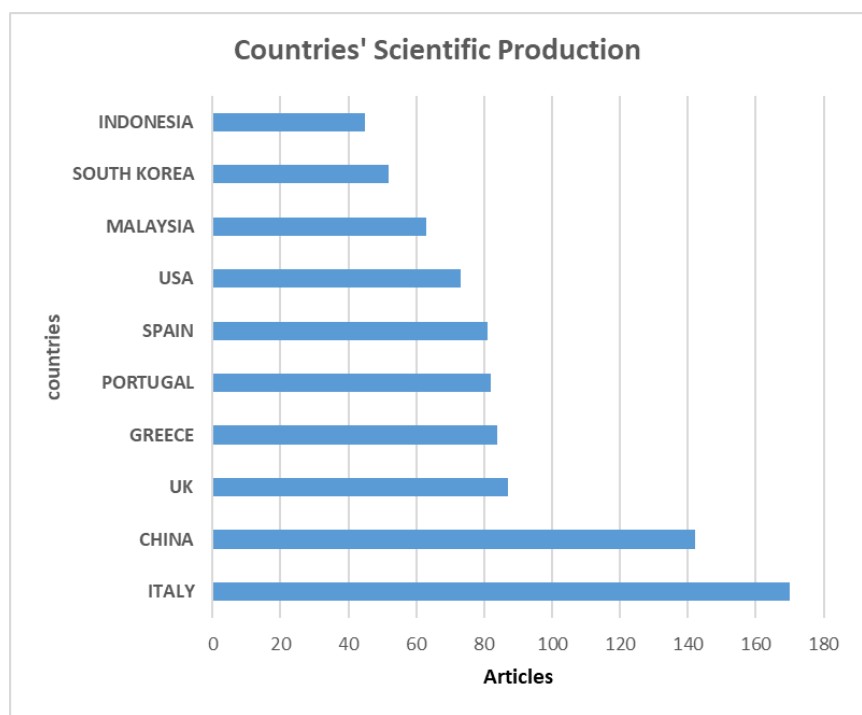

**Figure 4.** Top 10 most productive countries.

A nuanced examination of the economic classification of countries involved in AR tourism research unveils a heterogeneous landscape. Advanced economies such as Italy, the UK, and the USA are undoubtedly prominent players; however, the substantial contributions from emerging economies such as China, Malaysia, and Indonesia underscore the inclusive nature of the field. This observation suggests that AR tourism research transcends economic boundaries, fostering participation from nations across a broad spectrum of income levels and developmental stages.

A thorough analysis of the most cited countries in AR tourism research, as depicted in Figure 5, highlights a diverse landscape that transcends geographic borders. The United Kingdom's leading with 1172 citations demonstrates the nation's commitment to producing high-quality research in AR tourism. South Korea having 1084 citations reflects the country's rapid technological advancements, and its keen interest in the development of innovative applications for AR in tourism. Spain and Italy contributing 995 and 856 citations, respectively, underscores the importance of AR in preserving and promoting cultural heritage within the tourism sector. Both countries have a rich history and a plethora of tourist attractions, which has driven the need for groundbreaking research in AR, to enhance visitor experiences and boost tourism. The USA having 815 citations displays the nation's status as a global leader in technology and innovation. The significant citation count is a testament to the quality of research produced by excellent universities and research centers across the country.

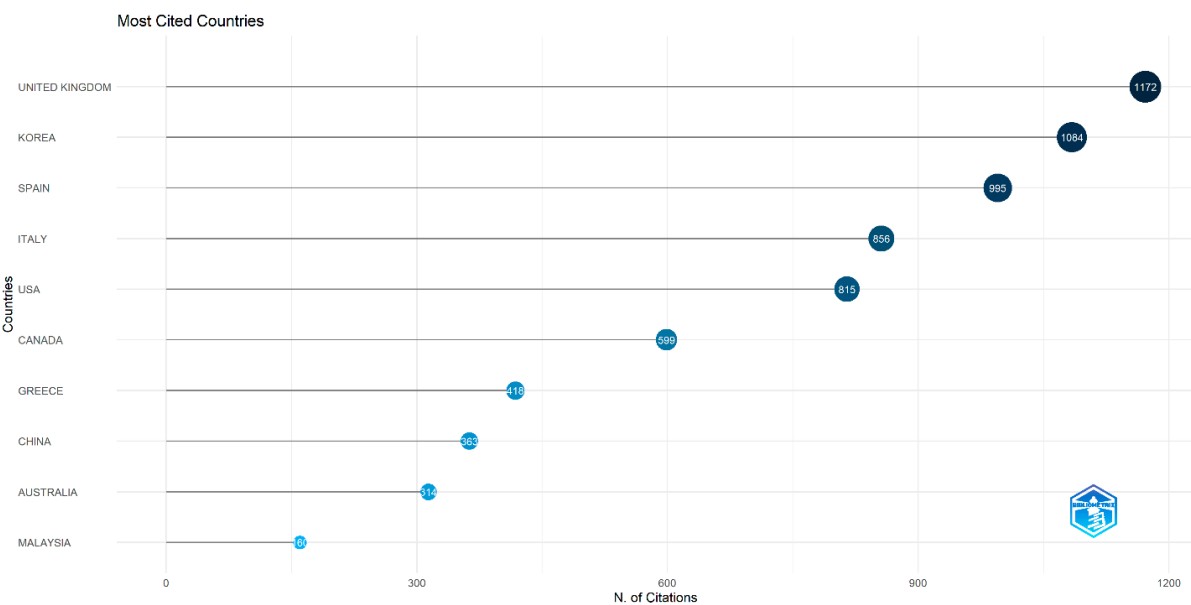

**Figure 5.** Top 10 most cited countries.

### 4.2. Emerging Topics of AR in Tourism

Co-word analysis was conducted on the 861 articles published about the application of AR in tourism using the authors' keywords to study the network of relationships between these themes, based on the co-occurrence of keywords (see Figure 6), and to discover the emergence of themes over time (see Figure 7).

Figure 6 provides a network visualization of the conceptual structure of the most commonly occurring concepts in AR research within the tourism industry. Each node in the network diagram represents a specific keyword, while the links between nodes illustrate the frequency of co-occurrence of these keywords. The size of each node corresponds to the frequency of occurrence of the respective keyword, while the thickness of the links represents the strength of the association between the keywords. To facilitate interpretation, each thematic cluster in the network is color-coded, highlighting the most salient topics (nodes) and their interrelationships (links). It should be noted that the default settings and graphical parameters of the "Biblioshiny" tool were utilized throughout our bibliometric analysis. As a result, a minimum node size was maintained, to ensure visual clarity, regardless of variation in the frequencies of keyword occurrence.

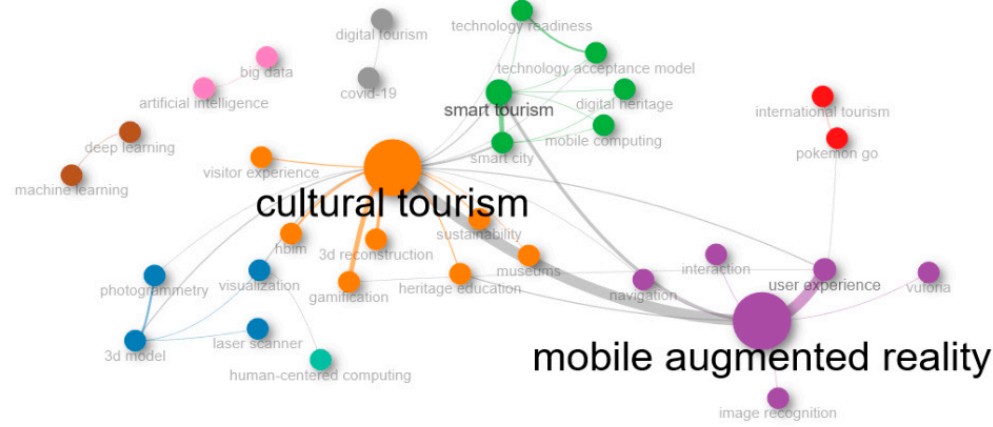

**Figure 6.** Keyword co-occurrence network based on author's keywords.

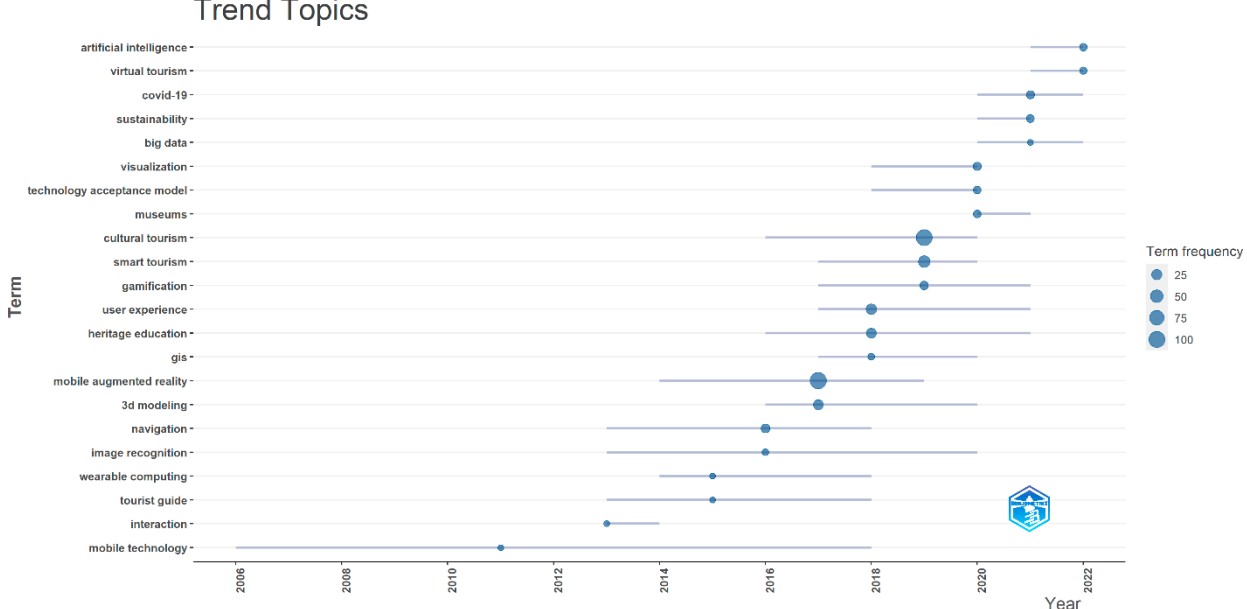

**Figure 7.** The occurrence of keywords and their frequency over time (years) based on the authors' keywords.

Our study has identified nine thematic clusters. Of these clusters, the two most significant are 'mobile augmented reality' (MAR) and 'cultural tourism' (CT), which are represented by the purple and orange colors, respectively. The MAR node is strongly associated with other nodes in the same cluster, such as 'image recognition', 'user experience', 'navigation', 'interaction', and 'Vuforia', indicating the centrality of MAR in AR research within the tourism industry. Similarly, the CT node is linked to 'visitor experience', 'sustainability', '3D reconstruction', 'gamification', 'heritage education', and 'museums', reflecting cultural tourism's significance in the field. The strong connection between MAR and CT in the network analysis, and their shared sub-themes suggest that cultural tourism presents a promising area for the application of MAR, to enrich the tourist experience through interactive navigation, virtual reconstructions, gamification, and heritage education. The 'smart tourism' (ST) node represents the third, green thematic cluster, which is closely linked to the CT node. The ST cluster includes nodes such as 'smart city', 'digital heritage', 'mobile computing', 'technology readiness', and 'technology acceptance model' (TAM). These nodes are also interconnected with the CT cluster node. This suggests the integration of MAR with smart city technologies, to enhance cultural tourism experiences [25]. TAM and technology readiness are two crucial nodes in the ST cluster, highlighting the significance of understanding tourists' acceptance of new technologies, and their readiness to adopt them for the successful implementation of smart 4.0 tourism solutions that incorporate AR technology [26]. The '3D visualization' cluster, identified as the fourth, blue cluster, comprises four nodes, including '3D model', 'photogrammetry', 'laser scanner', and 'visualization'. The nodes are interlinked with the CT cluster, indicating the potential of 3D visualization techniques to capture and generate 3D models of cultural heritage sites, thereby creating immersive AR experiences in tourism settings [27]. Additionally, the fifth, sky-blue cluster, which is represented by the single 'human-centered computing' node, is also linked to the 3D visualization cluster, emphasizing the significance of considering human aspects when designing and implementing AR systems in the tourism industry [28]. The sixth, red thematic cluster referred to as the 'gamification' cluster consists of two nodes: 'international tourism' and 'Pokémon Go'. The latter is linked to the MAR cluster through the 'user experience' node, suggesting the potential of using AR gaming applications such as Pokémon Go to attract international tourists and enhance their overall tourism experience [29]. The final three clusters in the network do not share any connections with the other identified

clusters. The first, brown cluster is composed of two nodes, namely 'machine learning' and 'deep learning'. The second, pink cluster consists of 'artificial intelligence' (IA) and 'big data', while the last, gray cluster includes 'COVID-19' and 'digital tourism'. It can be inferred that IA, machine learning, and virtual tourism in the post COVID-19 era may represent niche areas of research that have not been thoroughly explored in connection with AR research in tourism [8,30].

Figure 7 illustrates the evolution of AR research themes in tourism over time. Together with Figure 2, which displays the annual distribution of published papers, the 2002–2022 timeframe has been divided into five distinct phases.

The first phase, known as the Pioneering Phase (2002–2006), saw a small group of early visionaries exploring the potential of AR in the tourism industry, with a particular focus on mobile technology that would become increasingly important in subsequent years [31]. The second phase, referred to as the Mobile Engagement Era (2006–2013), emphasizes the growing importance of mobile technology, which facilitated tourists' access to augmented reality experiences, using handheld devices. During this phase, a focus was placed on developing engaging and interactive AR experiences for tourists [32]. The third phase, known as the Technological Convergence Period (2014–2017), is characterized by the emergence of wearable devices such as smart glasses, and the integration of AR with advanced navigation technologies such as GIS, as well as image-processing techniques such as 3D modeling and image recognition [33]. Heritage education and museums also gained prominence during this phase [34], coinciding with the growth of MAR, allowing visitors to better understand and appreciate cultural tourism experiences. The fourth phase is the User Experience Optimization Stage (2017–2019). At this stage, the focus shifted to enhancing the user experience, often through gamification techniques, and to developing the smart tourism concept [35]. Studies also investigated user acceptance of AR in tourism, seeking to understand the factors that influence user adoption and satisfaction [36]. Visualization techniques were further explored, to improve the overall user experience [37]. As a result, cultural tourism experienced significant growth in 2019, as shown in Figure 7. The final phase is the Adaptation and Integration Stage (2020–2022). The most recent stage in the development of AR in tourism is characterized by the use of big data to enhance AR experiences [38], and the growing importance of sustainability in the industry [8]. The COVID-19 pandemic led to an acceleration in the adoption of AR for virtual tourism [39], offering new possibilities for survival and recovery in the sector [40]. Furthermore, artificial intelligence has increasingly been integrated into AR applications, creating more personalized and dynamic experiences for tourists [41].

### *4.3. Most Utilized AR Design Components in Tourism Settings*

To address our third research question, we followed the PRISMA framework to systematically identify and select 57 articles for content analysis (see Figure 1). These articles were published between 2013 and 2022, with 3 (5.26%) published in 2013, 2 (3.50%) in 2014, 3 (5.26%) in 2015, 3 (5.26%) in 2016, 6 (10.52%) in 2017, 13 (22.80%) in 2018, 10 (17.54%) in 2019, 6 (10.52%) in 2020, 6 (10.52%) in 2021, and 5 (8.77%) in 2022. These results suggest an increasing trend in the number of publications on AR in tourism over the years, with a significant increase in 2018. Concerning the types of articles, 19 papers (33.34%) were published in conference proceedings, 35 papers were journal articles (61.40%), and 3 articles were published as book chapters (5.26%). All 57 articles were written in English. The geographical distribution of the 57 articles, based on the first author's affiliation, spans four continents. Thirty-one articles are from Europe (54.39%), with countries such as the United Kingdom, Spain, Greece, France, Portugal, and others. Twenty-two articles are from Asia (38.60%), with research conducted in China, Japan, South Korea, Taiwan, the Philippines, and elsewhere. Three articles are from Africa (5.26%), with a focus on Morocco; and one article is from South America (1.75%), with a feature on Colombia. These results underscore the international traction of augmented reality in tourism, with a significant lead for Europe, followed by Asia.

### 4.3.1. AR Systems

A content analysis of the 57 articles revealed a variety of AR systems being researched in the context of tourism. As shown in Figure 8a, marker-based systems, which rely on specific visual cues, such as art gallery paintings [42], food menus [43], and museum artifacts [44] for the accurate tracking and rendering of virtual information, account for 18 articles, with 13 focused on image-based systems, and 5 on object-based systems. Marker-less systems, which do not require predefined markers for tracking, are more prominently represented in the literature, with 30 articles in total. Among them, 21 articles explore location-based systems (e.g., [45]), 8 delve into superimposition-based systems (e.g., [46]), and only 1 investigates projection-based systems [47]. Additionally, 9 articles examine hybrid systems that combine elements from both marker-based and marker-less approaches (e.g., [48]).

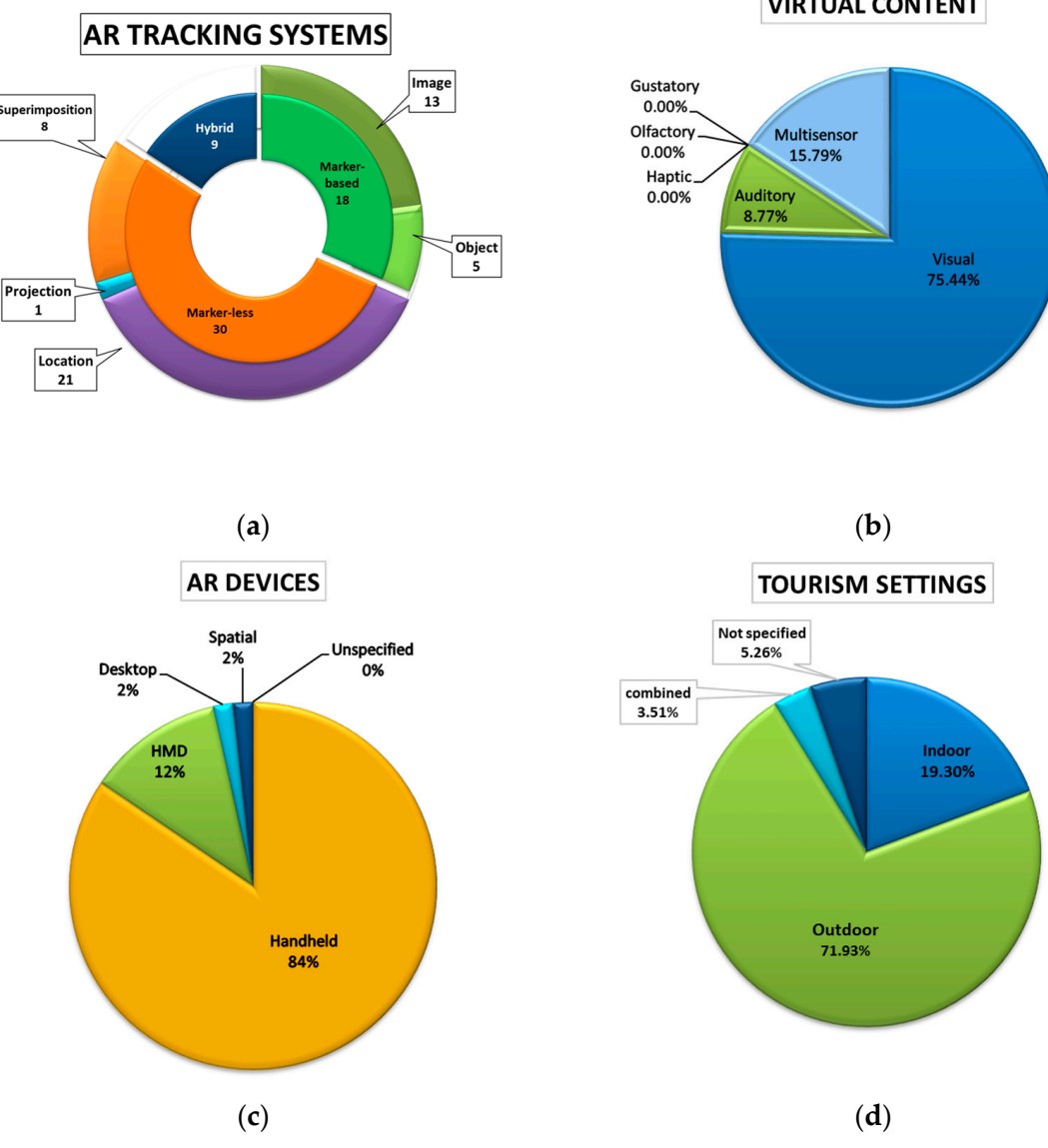

**Figure 8.** AR design elements. (**a**) AR tracking systems. (**b**) Virtual content overlaid by AR devices. (**c**) AR devices used to deploy AR experiences. (**d**) Tourism settings supported by AR technology.

### 4.3.2. AR Devices

Figure 8c demonstrates the range of devices that researchers and practitioners have been exploring to deliver AR experiences in the tourism industry. Handheld devices, espe-

cially smartphones and tablets, were the most commonly used, featuring in 49 of the articles (84%). This prevalence can be attributed to the widespread accessibility and portability of mobile devices, making them a popular choice for implementing AR experiences in tourism. Head-mounted displays (HMDs), particularly smart glasses and headsets, were utilized in 7 articles (12%), offering a more immersive experience for users. Desktop systems were employed in just 1 study (2%), while spatial devices, which enable users to interact with their surroundings using video projectors [47], also appeared in 1 article (2%). Notably, none of the articles left the device type unspecified. Furthermore, the total number of articles exceeds 57, as some studies report the use of more than one AR tool.

### 4.3.3. Tourism Settings

Figure 8b reveals the different tourism settings in which AR is being applied. The majority of the studies, with 41 articles (71.93%), focus on outdoor settings, highlighting the potential of AR to enhance the experience of tourists visiting natural and urban environments [49]. In contrast, 11 articles (19.30%) explore indoor settings, such as museums, galleries, and historical buildings, demonstrating the value of augmented reality in providing immersive experiences within confined spaces (e.g., [50]). Additionally, 2 articles (3.51%) investigate the combined use of AR technology in both indoor and outdoor environments, emphasizing the versatility of the technology [51,52]. It is noteworthy that 3 articles (5.26%) do not specify the tourism setting in their studies [43,53,54].

### 4.3.4. Virtual Content

The content analysis of the 57 articles, as shown in Figure 8d, reveals the various virtual content types employed in AR experiences for tourism. The most predominant type of content is visual, featured in 43 articles (75.44%), which aligns with the primary objective of augmented reality, to overlay digital information onto the physical world. Auditory content, although less common, was employed in 5 articles (8.77%), providing an additional layer of sensory experience for users. It is intriguing to observe that none of the papers focus on haptic, olfactory, or gustatory content individually. However, 9 studies (15.79%) have developed systems that incorporate these sensory experiences simultaneously, through multi-sensor systems. For example, the system presented by Rodrigues et al. [44] consists of an AR mobile application handling sight and hearing by recognizing and tracking museum artifacts, and a portable device enhancing the AR experience by stimulating touch, taste, and smell when connected to the user's mobile device.

### 4.3.5. Links between AR Tracking Systems and Tourism Settings

Table 6 presents the matrix of relationships between AR systems (rows) and tourism settings (columns), where they were employed to improve the tourist experience. Noticeable preferences for particular systems can be seen, depending on the environment. It is important to note that the total sum of articles exceeds 57, as numerous studies employ more than one system to accommodate various tourism settings. The values in parentheses correspond to the instances of the matrix.

In indoor settings, marker-based (6) and marker-less (5) systems exhibit a nearly equal distribution. Marker-based image (5) systems are the most prevalent, effectively enhancing visitor experiences by overlaying visual information onto images, making them well-suited for indoor environments such as museums and galleries [42,55]. Although marker-less systems are not as popular in indoor situations, location-based and superimposition-based applications have been found to be effective in improving user engagement in museums [56] and facilitating interactive experiences [57].

Outdoor settings demonstrate a clear preference for marker-less location-based applications (24). Such systems leverage the positions of users and landmarks to enrich tourism experiences, including navigation and open-site exploration [58–60]. Marker-less superimposition-based systems (10) have found more applications in outdoor settings than in indoor ones. This trend could be attributed to the growing demand for virtual tourism in

the post-COVID-19 era, as these applications provide immersive experiences allowing users to explore and interact with virtual monuments in real-life dimensions [61,62]. Interestingly, marker-based image (8) and object (6) systems have also seen more frequent use in outdoor settings. This suggests that these approaches can effectively enhance visitor experiences in outdoor locations [63].

**Table 6.** Interrelationships between AR tracking systems and tourism settings.

| | Indoor | Outdoor | Combined Settings | Not Specified |
|---|---|---|---|---|
| Marker-based image | 5 [42,55,64–66] | 8 [62,63,67–72] | 2 [51,52] | 3 [43,53,54] |
| Marker-based object | 1 [44] | 6 [48,73–77] | 1 [51] | 0 |
| Marker-less location | 2 [50,56] | 24 [28,35,45,49,58–61,70,72,75,78–90] | 1 [52] | 0 |
| Marker-less projection | 1 [47] | 0 | 0 | 0 |
| Marker-less superimposition | 2 [57,91] | 10 [46,48,49,61,62,92–96] | 0 | 0 |

In combined indoor/outdoor settings, there is a more even distribution of AR system usage, with the marker-based image (2), marker-based object (1), and marker-less, location (1) systems being implemented. The studies in question employ a combination of systems to support different environments [51,52]. Finally, in cases where the setting is not specified, marker-based image (3) systems emerge as the most common choice. This may indicate the flexibility of certain AR systems in overcoming the limitations imposed by diverse tourism settings.

In conclusion, this analysis reveals that the choice of AR systems in tourism research is closely linked to the specific setting in which they are applied. Marker-based image systems are favored in indoor environments, while outdoor settings predominantly utilize marker-less, location-based systems. The ability of certain AR systems to function effectively in combined settings underscores their adaptability, and potential for broader applicability within the tourism sector.

### 4.3.6. Links between AR Devices and Virtual Content

Table 7 illustrates the matrix of relations between AR devices (rows) and the virtual content (columns) they produce, examining how these elements work together to improve user experience. It is important to note that the total sum of rows and columns is 59, as some projects have utilized multiple tools to provide multisensory content. For instance, ref. [85] presented two location-based AR systems that employed both smartphones and smart glasses, enabling users to access audio and visual information when arriving at a point of interest.

**Table 7.** Interrelationships between AR devices and virtual content.

| | Visual | Auditory | Haptic | Olfactory | Gustatory | Multisensory |
|---|---|---|---|---|---|---|
| Handheld | 39 [28,43,45,48–51,53–55,58,59,61–65,67,68,71–76,79–81,83,84,86–90,93–96] | 2 [35,60] | 0 | 0 | 0 | 8 [44,46,52,69,70,77,85,92] |
| HMD | 3 [42,57,66] | 3 [56,78,82] | 0 | 0 | 0 | 2 [85,91] |

**Table 7.** *Cont.*

|  | Visual | Auditory | Haptic | Olfactory | Gustatory | Multisensory |
|---|---|---|---|---|---|---|
| Desktop | 1 [80] | 0 | 0 | 0 | 0 | 0 |
| Spatial | 1 [47] | 0 | 0 | 0 | 0 | 0 |

The matrix indicates that users are primarily attracted to handheld devices, such as smartphones, tablets, or even portable telescopes, as demonstrated by [80], for creating various types of virtual content. A substantial emphasis is placed on visual (39) and multisensory (8) content. For example, users were able to employ their smartphones to superimpose 3D models of historical monuments in accurate dimensions [51,61], or navigate to points of interest using annotations or virtual paths projected onto the real world [28,49,83]. Tourists could also generate 3D reconstructions of historical sites that are submerged underwater [48]. These results are consistent with the emergence of mobile AR as a central topic in AR tourism research.

HMDs, on the other hand, are predominantly utilized for visual content (3), with smart glasses enabling users to interact with 3D animations [57], or enhance learning experiences at art galleries [66]. Similarly, HMDs are employed for auditory content (3), with headsets serving as audio AR tour guides in indoor museums [56] and outdoor environments [78,82]. HMDs have also been used for multisensory content (2), incorporating smart glasses with audio output, to facilitate exploration and navigation of historic sites [85]. This evidence demonstrates that researchers are actively pursuing more immersive AR experiences through HMDs, even though handheld devices currently dominate the research landscape.

Regarding desktop and spatial devices, both have a very limited representation, with each being used only once for generating visual content. This could be attributed to the fact that desktop and spatial devices, such as AR projectors, are less practical, and have limited sensor capacities for AR application in tourism. As a result, researchers and practitioners may be less inclined to explore these devices in the context of AR tourism for enhancing user experiences, instead focusing on more portable and versatile options such as handheld devices and HMDs.

## 5. Future Directions

Based on the above findings, our thoughts on the future directions of AR in tourism are as follows:

- **Artificial intelligence (AI) and machine learning**: a number of research projects have begun to explore the potential of artificial intelligence and machine learning techniques, to design more personalized and dynamic tourism experiences [45]. The integration of deep-learning algorithms would allow the customization of AR content to suit individual demands, and create engaging experiences according to user preferences, behaviors, and patterns. Future works could consider exploring the integration of generative AI models, to enhance user interactions through natural language processing (NLP), and automate the process of creating AR content. By leveraging the power of generative AI, AR applications could provide more intuitive and seamless experiences, while reducing the time and resources needed for content creation.
- **Big data**: according to Rezaee et al. [86], one of the challenges in adopting AR for providing the right services to users is the lack of necessary data resources, despite the increasing amount of spatial information generated by people on a daily basis. The utilization of big data analytics could help enhance AR experiences, by providing real-time information on tourists, destinations, and other relevant factors. This could improve decision-making, user recommendations, and the overall quality of AR content for tourists.

- **Focus on sustainability:** with sustainability concerns on the rise, AR applications could play a crucial role in educating tourists about the significance of preserving natural and cultural heritage, and minimizing their environmental impact. By using superimposition-based systems, for instance, AR could also provide virtual experiences to explore historical monuments, without the need for physical presence, potentially reducing the sustainability issues related to the extensive mobility of large groups of tourists in traditional tourism [2].

- **Internet of Things IoT:** the findings of the bibliometric analysis suggest a growing focus on smart tourism research in the context of smart cities and AR. As IoT technologies continue to develop, future research should consider the integration of AR with IoT to provide real-time information on public transportation [97], events, and other services, making it easier for tourists to navigate and explore their surroundings and, consequently, expanding the role of AR in smart tourism.

- **Wearable AR:** future studies should increase the focus on wearable displays. A majority (84.84%) of the systematically reviewed studies employed AR experiences on handheld devices, whereas only a small portion (12.07%) utilized HMDs for AR technology implementation. This suggests a growing emphasis on HMDs to offer a more immersive experience for users. As a result, we might anticipate the emergence of comfortable, low-cost, and user-friendly wearable devices, such as AR glasses [98] and headsets that could improve and expand user adoption of AR technology.

- **Multimodal AR:** as revealed earlier, research on the use of AR technology to support haptic, olfactory, and gustatory content in tourism is currently lacking. This may be due to various factors, such as technological limitations, accessibility, and user acceptance. To address this gap, future research could explore the development, implementation, and impact of these sensory experiences in AR applications for tourism. By incorporating multiple sensory modalities, multimodal AR experiences have the potential to provide more immersive experiences for tourists.

- **Virtual and remote tourism:** in light of the post-COVID-19 era, there is a growing demand for the development of virtual and remote tourism experiences [99]. Future works could address this gap, by providing tourists with immersive and interactive AR experiences that allow them to explore and learn about destinations in the absence of physical travel.

- **UX optimization:** future AR studies in tourism will likely focus more on advancing the optimization of user experiences. This may include expanding the use of gamification techniques, improved navigation for both indoor and outdoor settings, and customizable user interfaces to accommodate the specific needs of users with disabilities. Moreover, as highlighted by this study, AR in tourism is an international trend (see Figure 3). Therefore, future AR applications should support multicultural experiences, to cater to a diverse range of tourists with different cultural backgrounds.

## 6. Conclusions

This study presents a comprehensive literature review of AR research in the tourism industry from 2002 to 2022, conducting bibliometric and systematic content analysis to reveal the evolution and application contexts of AR technology to support tourism, as well as identifying the AR design elements most used to enhance the tourist experience. Figure 9 provides a summary of the methods used, and the key findings obtained from the review. Findings indicate a growing interest in AR for tourism, with a significant and steady increase in publications since 2008. Europe and Asia are leading in research efforts.

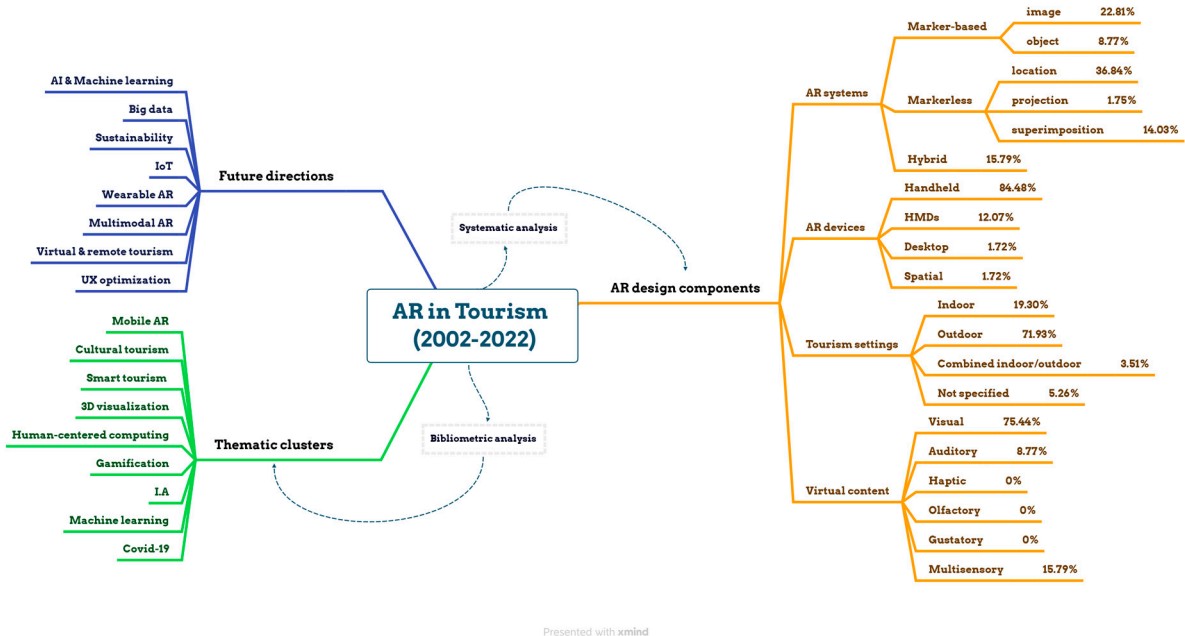

**Figure 9.** Overview of the methods and the main findings of the study.

The bibliometric analysis of 861 published papers identified mobile AR, cultural tourism, and smart tourism as the most prominent themes in AR research within tourism. The study also outlines the evolution of AR research themes in tourism, emphasizing the shift from early exploration to technological convergence, focusing on user experience optimization, and finally adaptation and integration. As the field continues to evolve, emerging topics include the integration of AI, machine learning, and big data to create more personalized and dynamic experiences for tourists. Sustainability and virtual tourism have emerged as well, in response to the changing landscape caused by the COVID-19 pandemic, and the growing demand for sustainable practices in the tourism industry.

The systematic content analysis of 57 selected articles demonstrates that AR marker-less systems are more prominent than marker-based systems, with location-based systems being the most explored. Handheld devices, such as smartphones and tablets, dominate the AR devices used in tourism, due to their accessibility and portability. The majority of analyzed papers focus on outdoor tourism settings. Indoor settings such as museums, galleries, and historical buildings also benefit from AR applications. Some studies explored AR's combined use in indoor and outdoor environments, showcasing the technology's versatility. Visual content is the most prevalent in AR experiences for tourism, followed by auditory content. While haptic, olfactory, and gustatory content types are not individually explored, some studies have developed multi-sensor systems that incorporate these sensory experiences simultaneously, enhancing the overall AR experience.

Future directions for AR research in tourism emphasize the exploration of multimodal AR experiences, the integration of AI and big data, sustainability support, and implementation of virtual-remote tourism. Moreover, they emphasize the importance of exploring more wearable AR applications, and optimizing user experience, to meet the needs of a wide range of tourists and enable multicultural experiences.

Theoretical and practical implications can be drawn from our review, to initiate new research and discussions on tourism and AR at the academic and industrial levels. In theory, this paper is a basic resource for AR and tourism researchers. It provides a comprehensive review of current research literature, including bibliography analysis and systematic content analysis, highlights key themes, identifies new trends, and examines the development of AR technologies in tourism between 2002 and 2022. In other words, scholars can use this article to point out where, when, and how AR is used to support tourism. This information can help researchers to identify gaps in knowledge, and future research avenues. We also

point out that the combination of bibliometric and systematic analysis is an effective way to gain a holistic understanding of the structure of emerging disciplines. Future review articles might follow the same method [4].

From a practical standpoint, this paper serves as a valuable resource for various stakeholders in the tourism industry who are interested in improving the overall tourist experience through the latest Industry 4.0 technologies, particularly AR technology. The results of the systematic content analysis identify the most commonly used AR design elements, and discuss their implementation to enhance user experience, which can guide tourism product developers to introduce innovative and robust AR-based applications, thereby improving consumer satisfaction, and reducing the technological challenges forced by the industry. Furthermore, this review could also be informative to policymakers and tourism authorities regarding the potential benefits of integrating AR into tourism propositions, guiding the formulation of supportive policies and investments in AR technologies, thereby contributing to the growth and sustainability of the tourism sector.

Despite the comprehensive methodology employed in this study, some limitations should be acknowledged. Firstly, the study only focused on articles published in English, which could have resulted in a language bias, thus potentially omitting relevant research published in other languages. Future research could extend the scope of the review to include literature in other languages to provide a more holistic understanding of the application of augmented reality in the tourism industry. Secondly, the search strategy and data collection were restricted to two major databases, Scopus and Web of Science. Although these databases cover a substantial proportion of the scientific literature, there might be relevant studies published in other databases, or gray literature sources that were not captured in this review. Future research could expand the search to additional databases and sources, to minimize the risk of missing relevant publications. Lastly, this study employed bibliometric and systematic content analysis to explore the use of AR in the tourism industry. While these methods provide valuable insights, they are not without limitations. Co-word analysis relies on keyword counts, which can be influenced by various factors such as author preferences and evolving terminology. Furthermore, systematic content analysis is a subjective method that relies on the interpretation of researchers, which may introduce researcher bias. Future work could consider employing complementary methods, such as meta-analysis, to validate and enhance the findings. It is essential to acknowledge that this study was not registered, and the protocol used in the study was not archived. Given the utilization of a dual approach that combines bibliometric analysis and systematic review, it was not possible to register the study prior to its launch, and it presented unique challenges in terms of alignment with existing registration frameworks. Nonetheless, it is important to note that this study follows the PRISMA 2020 statement guidelines for systematic reviews, and employs best practices in bibliometric analysis.

**Supplementary Materials:** The following supporting information can be downloaded at: https://www.mdpi.com/article/10.3390/mti7070064/s1; Table S1: Summary of the studies (*n* = 57) included in the systematic review; File S1: PRISMA_2020_checklist.

**Author Contributions:** Conceptualization, M.Z. and H.K.; methodology, M.Z. and H.K.; formal analysis, M.Z. and H.K.; investigation, M.Z. and H.K.; writing—original draft preparation, M.Z. and H.K.; writing—review and editing, M.Z., H.K., F.Z.K. and A.A.; visualization, M.Z. and H.K.; supervision, F.Z.K., A.A. and K.A.Z.; project administration, A.A. All authors have read and agreed to the published version of the manuscript.

**Funding:** This research received no external funding.

**Institutional Review Board Statement:** Not applicable.

**Informed Consent Statement:** Not applicable.

**Data Availability Statement:** Not applicable.

**Conflicts of Interest:** The authors declare no conflict of interest.

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
