# Peer review of "From Earlier Exploration to Advanced Applications: Bibliometric and Systematic Review of Augmented Reality in the Tourism Industry (2002–2022)"

_mti, doi:10.3390/mti7070064_

Round 1

Reviewer 1 Report

This paper has provided an interesting topic on the field of AR in Tourism Industry. Although, it is a reviewed article but it also is a well-organized article.

Figure 1 is a good framework for understanding how the process of the systemic review. But, the Table 1 why concern about the language in ENGLISH. I would like to check that “is any research article in international journal does not use English?”  So, that is not necessary.

Most contents and results are so fruitful, but the contents are so many tables and figures. Which make reader sometime confuse and difficult read. For example, Fig.5 is not really important but showed.

All the references should be double checked. Because all of them have not following the format of journal.

Totally, I like this paper but too many Tables and Figures have made the reading difficult, it is better if simple if cleanly. 

no

Reviewer 2 Report

The paper's title and objective are easily understandable. The abstract provides valuable information about the article's content upon first glance. Nevertheless, it is worth noting that Table 1 indicates that addressing RQ3 is treated as an inclusion and exclusion criterion (as further elaborated in sub-chapter 4.3). Therefore, I recommend that the authors provide inside the abstract a clearer explanation of how a significant portion of their study relies on the analysis of only 57 articles. Even the title could be reformulated in order to clarify this issue, but for the abstract is a must. Reference list is relevant, but I will suggest later how this can be improved. The suggestion to enhance the reference list should be considered more as a minor revision recommendation.

The Introduction part clarifies what is already known about this topic. However, I suggest the authors to add some seminal works from that period (not only the systematic review article from 2021 – source 4) in order to give a better weight to the affirmation that “Researchers and tourism practitioners have recognized AR's potential in tourism since 2000”.

The objectives of the research are very clearly defined.

It is unclear for me if the bibliometric analysis was conducted on 849 articles or 861 records screened (as indicated in Figure 1).

Another suggestion to the authors is to give more details about the 160 records not retrieved.

Tables 2-5 offer valuable insights into the methodology, providing a detailed explanation that is likely to encourage future researchers to incorporate and acknowledge this article's methodology to some extent. However, for Table 3 is not very clear why having two distinct categories that might be merged (Unclear environment and Not specified). I saw later in the article (inside the Figure 9) that Unclear environment is not present. As I mentioned the Figure 9, in my opinion is better to change Figure 9 to Table 6.

I highly value the information presented in lines 253-259, as it is uncommon to find such comprehensive details in systematic literature review articles.

While Vuforia serves as a free software development kit for Mobile Augmented Reality, I have significant doubts regarding the nearly equal frequency of co-occurrences between the keyword "Vuforia" and the keyword "museum/museums." This aspect necessitates a thorough explanation or comment within this section of the article.

Congratulations for defining and explaining the five periods of the the annual distribution of published papers in the 2002-2022 timeframe.

As the sub-chapter 4.3 is an important part of the paper, I suggest the authors to make an extra-effort and introduce all 57 articles as references. For example, in lines 442, 443 and 445, the authors used e.g. and sending to references 41, 42 and 44. Why not creating a new table or changing the tables (now defined as Figure 9 and Figure 10) with including in this new tables the reference numbers to the articles that are mentioned now only as simple numbers. For example, I agree that for Marker-less location & Outdoor list of articles will be challenging and a consistent extra-effort will be needed to add 24 references numbers and not indicating that are 24 articles in this category. But for other categories from Figure 9 and Figure 10 (future tables) the effort will not be so consistent.

The sections on Future Directions and Conclusions exhibit exceptional writing and abound with invaluable insights.

Another notable aspect of this paper is the presence of Figure 11, which serves as an additional strong point.

The paper has received a highly positive evaluation overall, and implementing the suggested changes will greatly enhance its impact upon publication.

Reviewer 3 Report

The paper reports the results of a bibliographic and systematic reviews of augmented reality research and applications in the tourism industry, covering 278 articles from the past 21 years. This is a valid and ever-green field of research. 

The first part (bibliographic review) gives a good glimpse of how the research on AR developed in the field of tourism, the other (systematic review) maps the AR applications in tourism according to respective technologies and content types. The authors follow the PRISMA 2020 principles.

The results are presented in a clear way and recommendations towards future research directions are drawn.

There are, however, some issues that have to be addressed before the paper is accepted for publication:

1. The scope of the review is not precisely defined. It is not clear whether it covers the instances of research on augmented reality in the tourism industry or the applications of augmented reality in the tourism industry, or both (which I assume is the case). This should be made clear in the abstract and in the introduction.

2. The analysis of the prior work (section 2) is too limited. There is plenty of relevant work that has not been mentioned, including surveys exactly on the same topic [e.g. https://www.sciencedirect.com/science/article/pii/S0261517722000474 ], surveys of AR apps (including those applied to tourism) [e.g. https://ertr-ojs-tamu.tdl.org/ertr/article/view/559 ], surveys of tourism-dedicated apps (including those providing AR) [e.g. https://www.academia.edu/download/41839812/400.pdf ].

3. It is not clearly explained that different samples are used for bibliographic analysis and systematic review. 

4. There is either a methodological error in the bibliographic analysis or an error in Fig. 1, according to which the review was based on 1206 records including 345 duplicates. Obviously, duplicates should not have been considered in the bibliographic analysis. 

5. There is an issue with the structure of the paper. The general results of the review (shown in Fig. 11) should not be reported in section 6. Conclusion (which should recapitulate the main findings but not introduce new content), but in section 4. Findings and discussion.

Round 2

Reviewer 1 Report

This paper has revised most suggested points.

It is shitable to be considered to be published.

Reviewer 3 Report

Thank you for the corrections. All my concerns have been addressed. I have no new remarks.